# The First Use of a Midline Catheter in Outpatient Pain Management

**DOI:** 10.3390/healthcare12080856

**Published:** 2024-04-18

**Authors:** Kinga Olczyk-Miiller, Maciej Latos, Dariusz Kosson, Marcin Kołacz, Robert Hadzik

**Affiliations:** 11st Department of Anaesthesiology and Intensive Care, Medical University of Warsaw, 4 Lindleya Str., 02-005 Warsaw, Poland; 2Department of Anaesthesiology and Intensive Care Education, Medical University of Warsaw, 4 Oczki Str., 02-007 Warsaw, Polanddariusz.kosson@wum.edu.pl (D.K.)

**Keywords:** midline catheters, lidocaine, chronic pain treatment, interventional pain management, neuropathic pain

## Abstract

Midline catheters (MCs) are used to deliver intravenous therapy lasting over 5 days to patients in hospitals. However, the constant development of home and outpatient care is challenging medical teams to provide effective and safe planned therapy to patients under such conditions. We describe the first time an MC was used in outpatient pain management in Poland. A 60-year-old man presented to the Pain Management Clinic with a history of RCC of the left kidney and lumbar back pain radiating to the left knee joint. The person whose case is described below had poor peripheral veins. He intravenously received lidocaine for 10 days via a midline catheter with a good response.

## 1. Introduction

In standard management, midline catheters (MCs) are used to deliver intravenous therapy lasting over 5 days to patients in hospitals [1]. However, the constant development of home and outpatient care is challenging medical teams to provide effective and safe planned therapy to patients under such conditions [1]. Notwithstanding the underlying cause of the disease (e.g., oncologic disorders), various types of analgesic treatment involving different routes of drug administration are more and more often being implemented. When it is necessary to administer drugs via the intravenous route, the treatment team must choose the appropriate vascular access. Numerous cannulations using SPCs (short peripheral catheters) can lead to difficulties in obtaining intravenous access, especially when treatment exceeds 4 days [1,2,3].

While examining various IV access options, it is important to consider the patient’s current needs [1]. In cases where long-term chemotherapy and/or parenteral nutrition are required, central access methods are chosen, such as IVADs (implanted vascular access devices), PICCs (peripherally inserted central catheters), or PICC ports [1,4]. Peripheral access, i.e., MCs or LPCs (long peripheral catheters), can also be used in outpatient and home care. MCs are recommended for expected intravenous therapy lasting 5–14 days [1]. ERPIUP (European recommendations on the proper indication and use of peripheral venous access devices) guidelines recommend the use of MCs for patients requiring intravenous therapy for as long as several weeks [5]. Midline catheters can be used for compatible peripheral infusions of fluids, antibiotics, and analgesics [1,5]. Insertion of this type of catheter requires appropriate training in ultrasound-guided cannulation and the use of a direct Seldinger technique (DST) or modified Seldinger technique (MST) [1,5]. Midline catheters are inserted mid-arm and placed in the veins of the upper limb: the basilic, cephalic, or brachial veins [5]. Currently, the scientific debate revolves around the proper location of the catheter tip, i.e., in the so-called axillary or “midclavicular” line [1,5]. In any case, the proper position of the catheter tip must be chosen in such a way as to prevent mechanical complications and thrombosis associated with the presence of a vascular catheter [1,5,6]. Unlike central access, the insertion of MCs does not require the confirmation of proper positioning on an X-ray, which facilitates the use of this type of access in outpatient practice [1]. An important aspect in favor of using MCs is the reduced pain associated with suturing the catheter and the lower risk of injury due to the cannula adhering to the skin [7]. The rate of complications during MC infusion described by authors in the literature varies and is in the range of 12.5% [8]. Nevertheless, it must be stressed that maintenance of this type of access requires observing the basic principles of infusion care [8]. These include taking care of the patency of the catheter, applying the principles of ANTT (aseptic non-touch technique), and taking care of the exit site of the catheter [1]. However, in addition to the many benefits of using MCs, they have their limitations.

Due to their peripheral nature, they should not be used to infuse chemically incompatible solutions with extreme pH (appropriate pH is 5–9) [1]. If such solutions must be administered, PICCs should be considered due to the risk of extravasation of highly irritating drugs or the development of thrombosis [9]. Outpatients who receive a series of scheduled intravenous infusions can benefit from the advantages of MCs, primarily by avoiding the insertion of numerous SPCs and the associated pain, discomfort, and potential complications [10].

We describe, for the first time, the use of an MC in outpatient pain management in Poland. The person whose case is described below had poor peripheral veins. He intravenously received lidocaine for 10 days via a midline catheter.

## 2. Case Report

A 60-year-old man presented to the Pain Management Clinic with lumbar back pain radiating to the left knee joint. He had a medical history of generalized papillary renal cell carcinoma of the left kidney, a left nephrectomy in January 2021, and secondary cancer of the abdominal lymph nodes. He had undergone steroid therapy, chemotherapy, immunotherapy, stereotactic radiotherapy (SBRT) of the retroperitoneal space, and radiotherapy for a metastatic lesion in the rib. There was no information on chronic diseases other than oncologic disorders.

The patient had been experiencing pain since 2021. Initially, he reported mainly localized lumbar spine pain, with occasional spreading left-sided pain at the level of the hip bone. Since 2021, he has been successfully treated with oral analgesics of oxycodone with naloxone (10–20 mg oxycodone and 5–10 mg naloxone every 12 h) in combination with gabapentin 3 × 300 mg. His pain was controlled and constant, with intermittent exacerbations responding well to ad hoc administration of oral, fast-acting oxycodone preparations. In early April 2023, the patient suffered a worsening of pain, and for this reason, he visited the Pain Management Clinic.

The patient’s pain characteristics were rated on the numerical rating scale (NRS), PainDETECT questionnaire (PDQ), and Central Sensitization Inventory (CSI). The NRS evaluates subjective pain intensity on a scale from 0 to 10 (where 0 means no pain, and 10 means the strongest pain). According to the proposal of Boonstra et al., NRS 1–3 means mild pain, NRS 4–6 means moderate pain, and NRS ≥ 7 means severe pain. The treatment is considered successful if the pain intensity is lowered to a level of 0–3 on the NRS scale [11].

PDQ is a screening tool that assesses the likelihood of a neuropathic pain component. Scoring results are positive (neuropathic pain component is likely (>90%), PDQ 18–38 pts.), negative (neuropathic component is unlikely (<15%), PDQ 0–11 pts.), or unclear (PDG 12–17 pts.) [12].

CSI is a self-reported screening instrument to help identify patients with central sensitivity syndrome (CSS), defined as increased responsiveness of nociceptive neurons in the central nervous system to their normal or subthreshold afferent input. CSS is suggested to increase the likelihood of pain becoming chronic [13,14].

CSI evaluates CSS on a scale of 0–100 pts. The CSS can be subclinical (CSI 0–29 pts.), mild (30–39 pts.), mediocre (40–49 pts.), severe (50–59 pts.), or extremal (60–100 pts.). A cutoff score of 40 pts. provides a clinically relevant guide to the possibility that a patient’s symptom presentation may indicate the presence of a CSS [15].

During the visit, the patient reported pain radiating down the inner side of his thigh to the level of the left knee joint. He rated the constant daily intensity of his complaints at NRS 3–4, and during pain attacks, he evaluated his NRS as up to 8. The pain had certain features of neuropathic pain. The patient reported strongly expressed burning, tingling, and occasional attacks of pain in the form of electric shock in the innervation of the dorsal branches of L3 on the left side. On clinical examination, sensory disturbances in this area in the form of allodynia and hypoesthesia were found. His PDQ result was 12 points, while the CSI was 37 pts.

An imaging study (MRI of the lumbosacral spine) showed a 3.5 mm intradural lesion at Th 12 in the terminal part of the medullary cone on the left side—most likely a metastatic lesion. There was an extradural, paravertebral, and para-aortic conglomerate of lesions also involving the iliopsoas muscle and forming polycyclic masses extending for about 138 mm. These were lymph nodes and tumor metastases. At the L3–L4 level, there were visible metastases into the left intervertebral foramen with displacement and compression of the nerve roots. On the left side, there were spinal nerves among the conglomerate of the described lesions (Figure 1).

The patient received intensified analgesic treatment. The base dose of the oxycodone and naloxone combination was gradually raised to a daily dose of 120 mg + 60 mg. Subsequently, extended-release oxycodone formulations were added. Gabapentin treatment was converted to a supply of pregabalin at a dose of 2 × 150 mg and subsequently 2 × 300 mg. Duloxetine (90 mg/d) and periodic steroid therapy were included, achieving a satisfactory analgesic effect for several more months.

In November 2023, despite the use of pharmacotherapy, the severity of complaints rose to NRS 8–9. The extent of pain and its type were the same as reported earlier. On imaging (MRI of the lumbosacral spine), the spread of secondary metastases into the intervertebral orifices did not show significant dynamics. The dimensions of the intradural nodule at the Th12 level did not change. There was a slight progression of secondary tissue changes in the left iliopsoas and para-aortic space. The bone parts did not have any suspicious focal lesions. In consultation with neurosurgeons, the process of qualifying the patient for SCS (spinal cord stimulation) or ITDD (intrathecal drug delivery) was initiated. Given the patient’s strongly expressed complaints of neuropathic pain, the decision was made to start a cycle of 10 intravenous lidocaine infusions at a dose of 4–5 mg/kg. These were administered daily from Monday to Friday in a procedure room setting at the local Pain Management Clinic, where monitoring of his vital signs (HR, NIBP, and SpO_2_) was conducted while administering the treatment and 15 min after completing the IV infusion.

Given the depletion of superficial vein resources in the patient, the anticipated duration of the therapy, and the very good cooperation with the patient, the decision was made to implant a midline catheter and use it in an outpatient setting. The MC was inserted under the conditions of the procedure room of the Outpatient Pain Management Clinic after obtaining written consent from the patient. The procedure was performed by an anesthesiologist employed by the clinic. While applying the maximal sterile procedure, a 4 Fr 20 cm MC was inserted under ultrasound guidance (linear probe) into the right basilic vein using a DST. Confirmation was obtained by free blood aspiration, flushable with 0.9% NaCl, and the position of the tip in the axillary line was checked by ultrasound. The catheter was attached using a sutureless system, and a transparent dressing was applied (Appendix A). The type of access was marked with a label indicating that an MC was inserted (Figure 2).

The patient received the necessary information on proper MC care. He was instructed to carry out daily activities as normal but to pay special attention to the area around the catheter insertion, i.e., tightness of the dressing, closed catheter clip, and reflux of blood in the extension tube. It was also instructed how the patient’s family could take care of the catheter maintenance between daily visits, including the correct way to disinfect the needle-free connector, the method of flushing, and the correct closure of the catheter. The use of vascular access other than IVAD or tunneled centrally inserted central catheters is not standard in home care in Poland. In the case described, observation and dressing changes (every 7 days) were performed daily by the Pain Management Clinic [1]. There were 10 scheduled intravenous infusions of lidocaine, and no adverse reactions related to its administration were reported. No features of infection in connection with the presence of the intravenous catheter were observed. Catheter patency was unproblematic. The patient did not report any difficulties due to the daily use of the cannula by the staff. He gave positive feedback on the way the catheter was implanted and the subsequent visits related to IV drug administration. The analgesic efficacy of the infusions was good. During the treatment cycle, the patient reported significant improvement (NRS decreased from 8–9 to 1–2), which allowed for a reduction to complete discontinuation of opioid treatment. After the 10th infusion, the MC was removed in the treatment room of the Pain Management Clinic. The patient’s satisfactory analgesic effect was maintained throughout the period of lidocaine infusion management and 10 days after infusion, which confirms its high efficacy in controlling neuropathic pain in patients under palliative care. After this time, the pain intensity began to gradually increase, forcing the patient to restart opioid therapy.

## 3. Discussion

In patients enrolled in the type of treatment described, a new SPC is inserted each day. Unfortunately, in quite a large group of patients, technical difficulties occur in the form of limited possibilities of establishing peripheral intravenous access. Obtaining further peripheral access is associated with discomfort, stress, pain, and decreased satisfaction with the treatment. The patient in question was typical of such cases due to a history of oncologic disease and intensive systemic treatment that had been administered for a few years. In this patient group, vein preservation is of particular importance [1]. Different pain management techniques are required for patients undergoing long-term oncologic and non-oncologic treatment. Not every patient has medium- or long-term intravenous access. This may be related to the nature of the planned therapy, organizational capabilities, medical contraindications, or the progression of the patient’s disease. However, MCs, PICCs, and PICC ports are not widely used in patients who do not have IVADs, despite the fact that they provide safe parenteral access in patients in whom intravenous symptom management or analgesic treatment is indicated [16]. Their use may facilitate IV symptom management outside the hospital and complement the traditional practice of subcutaneous administration [1,16]. In retrospective studies, vascular access events including infiltration, occlusion, and displacement were significantly more common in MCs than in other methods [17,18,19]. However, this may be related to the overall higher incidence of complications with peripheral access than with central access, especially when used for long-term therapy [10]. On the other hand, the complication rate associated with SPCs is unacceptably high, especially in patients exposed to multiple insertions over 4 days, and is approximately 36–70% [20,21]. MCs offer a comparable rate of device-related bloodstream infection to SPCs but with a significantly lower rate than PICCs and centrally inserted central catheters (CICC) [22]. For this reason, the Infusion Nurses Society (INS) recommends the use of MCs for 5–14 days of therapy and suggests considering central access (PICC) if extended therapy is necessary [1]. However, ERPIUP recommends using MCs for up to several weeks [5]. Home care should be planned and take into account the patient’s ability to cooperate and their motivation. Guidelines usually focus on full home care, which also includes the home administration of medication [1]. In our case, it was the patient’s responsibility to pay attention to potential complications that may arise while at home, such as bleeding from the catheter insertion site or blood reflux leading to occlusion. We recommend that if the patient is at home, they should always be able to make a phone call and get help if they have concerns. The INS recommends weighing up the benefits against the potential risks, which in our case supported the use of MC for therapy [1]. In the case described here, the use of lidocaine for 10 days combined with difficult intravenous access was a clear indication for the use of an MC, the popularity of which is steadily increasing in Poland. Lidocaine has a pH of about 6–7 and an osmolarity of about 300, so it falls into the “low risk” group of drugs for vein endothelial irritation, which can lead to phlebitis and thrombosis [23,24]. Thoughtful minimization of the risk of complications, explanation of the catheter maintenance regime to the patient and its application by staff, and cooperation with the patient are likely to avoid them in practice. The results of the studies available suggest that MCs can have a beneficial effect on the quality of care and patient satisfaction, and the catheter insertion procedure is well tolerated, which is also confirmed by the case we described [25].

## 4. Conclusions

The use of a midline catheter by the described patient proved to be easy and safe. It increased his comfort and saved time for the medical staff. It is reasonable to educate medical personnel about the potential uses of midline catheters and encourage their use. Studies are needed to assess the cost-effectiveness of the widespread use of MCs and their role in outpatient care. Together with the use of MC in home care, facilities should be developed to support patients in the prevention of complications and to solve problems when they occur, e.g., through staff being available on call 24/7.

## Figures and Tables

**Figure 1 healthcare-12-00856-f001:**
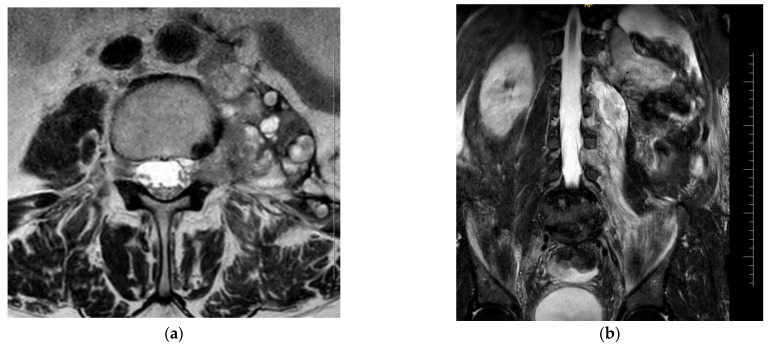
MRI of the lumbosacral spine: (**a**) metastases into the left intervertebral foramen with displacement and compression of the nerve roots at the L3–L4 level, cross-section; (**b**) an extradural, paravertebral, and para-aortic conglomerate involving the iliopsoas muscle, longitudinal section.

**Figure 2 healthcare-12-00856-f002:**
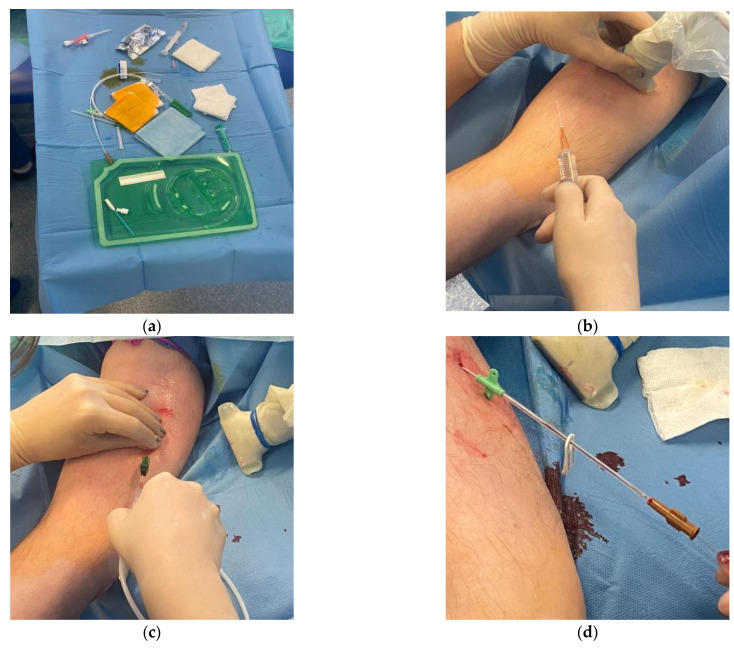
Insertion of the MC: (**a**) equipment; (**b**) ultrasound guidance; (**c**) insertion of the guide; (**d**) MC in place.

## Data Availability

Data are contained within the article.

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
