# Peer review of "The First Use of a Midline Catheter in Outpatient Pain Management"

_healthcare, 2024, doi:10.3390/healthcare12080856_

Round 1

Reviewer 1 Report

Comments and Suggestions for Authors

Cite any complications that could arise at home with a 5 day catheter.

What is the cost of home catheter?

Is there a maintenance routine for use of home catheter and do you need someone at home to monitor catheter delivery of the product?

Comments on the Quality of English Language

Good article.

After points addressed, may consider for publication

Reviewer 2 Report

Comments and Suggestions for Authors

Dear authors,

Your case report "The first use of an MC in outpatient pain managements" presents an interesting approach for applying midline catheters under specific conditions such as poor peripheral veins in an outpatient pain case with good results. Despite the contribution of this study to its field, I would like to comment on some concerns.

Major comments

1. Is there any additional care to apply this type of catheter in an outpatient individual? Once this is the first report, it is important to describe the similarities and differences of applying this system in ambulatory cases compared with bedridden patients.

2. "The patient received the necessary information on proper MC care" (line 134). What does it mean? Please describe all information sent to the patient. Was this totally or partially different from the information usually sent to bedridden patients?

3. "The analgesic efficacy of the infusions was good." (lines 141-142). How was this effect determined? Was the patient responding to a pain questionnaire or rating score? Please describe these findings in good detail.

4. You concluded that applying the midline catheter "increased his comfort" in the patient. How was this evaluated?

Minor comments

5. Please avoid abbreviations in the title. Correct "MC" for "midline catheters".

Reviewer 3 Report

Comments and Suggestions for Authors

In this study, Olczyk-Miiller et al reported a case. A 60-year-old man outpatient had lumbar back pain radiating to the left knee joint. For pain management, midline catheter (MC) was used for 10-day intravenous delivery of lidocaine with good response. The result of this study showed MCs could be easy and safe. In general, this case report is interesting, which suggests MC could have a beneficial effect on the quality of care and patient satisfaction. Below are several concerns and comments:

1. In lines 85-87 and 90-92, the authors described “He rated the constant daily intensity of his complaints at NRS 3-4, and during pain attacks evaluated his NRS as up to 8.” and “His PainDETECT questionnaire result was 12 points, while the Central Sensitization Index (CSI): 37 pts”. The author should explain what do these numbers mean about pain intensity.

2. In lines 97-100. The authors describe the nerves compression by tumor metastases, which might be the cause of lumbar back pain radiating to the left knee joint. It would be nice if these images could be provided.

3. The author mentioned that “during the treatment cycle, the patient reported significant improvement (NRS 1-2)”. It’s not clear when are the NRS measured, before, during or after the Lidocaine was infused. The time point of NRS is critical to evaluate whether the Lidocaine infusion management is a good strategy for pain management. It’s also important to know how long the analgesic effect lasts after the Lidocaine infusion and after the 10 Lidocaine infusion cycle was done.

4. Although Midline catheter appeared to be easy and safe in this case report, a greater number of cases should be included for evaluating the safety and efficacy of MC for outpatient usage, and the author should carefully discuss the potential risks of Midline catheter for outpatient usage, especially when outpatients are at home.

Round 2

Reviewer 2 Report

Comments and Suggestions for Authors

Dear authors,

Your case report "The first use of the Midline Catheter in outpatient pain management" presents an interesting approach for applying midline catheters under specific conditions such as poor peripheral veins in an outpatient pain case with good results. Thank you for addressing my previous comments.